# Gait Flexibility among Older Persons Significantly More Impaired in Fallers Than Non-Fallers—A Longitudinal Study

**DOI:** 10.3390/ijerph18137074

**Published:** 2021-07-02

**Authors:** Eva Ekvall Hansson, Elina Valkonen, Ulrika Olsson Möller, Yi Chen Lin, Måns Magnusson, Per-Anders Fransson

**Affiliations:** 1Department of Health Sciences, Lund University, 22100 Lund, Sweden; 2Helsinki University Hospital, 00280 Helsinki, Finland; elina.valkonen@hotmail.com; 3Department of Health Sciences, Kristianstad University, 29139 Kristianstad, Sweden; ulrika.olsson_moller@hkr.se; 4Alders Physiotherapy Clinic, Taipei, Taiwan; yichenlin217@gmail.com; 5Department of Clinical Sciences in Lund/ENT, Lund University, 22185 Lund, Sweden; mans.magnusson@med.lu.se (M.M.); per-anders.fransson@med.lu.se (P.-A.F.)

**Keywords:** balance, gait, falls, older people, postural balance

## Abstract

Gait disorders are a relevant factor for falls and possible to measure with wearable devices. If a wearable sensor can detect differences in gait parameters between fallers and non-fallers has not yet been studied. The aim of this study was to measure and compare gait parameters, vestibular function, and balance performance between fallers and non-fallers among a group of older persons. Participants were senior members (*n* = 101) of a Swedish non-profit gymnastic association. Gait parameters were obtained using an inertial measurement unit (IMU) that the participants wore on the leg while walking an obstacle course and on an even surface. Vestibular function was assessed by the Head-shake test, the Head impulse test, and the Dix–Hallpike maneuver. Balance was assessed by the Timed Up and Go, the Timed Up and Go manual, and the Timed Up and Go cognitive tests. Falls during the 12-month follow-up period were monitored using fall diaries. Forty-two persons (41%) had fallen during the 12-month follow-up. Fallers had more limited ability to vary their gait (gait flexibility) than non-fallers (*p* < 0.001). No other differences between fallers and non-fallers were found. The use of gait flexibility, captured by an IMU, seems better for identifying future fallers among healthy older persons than Timed Up and Go or Timed Up and Go combined with a cognitive or manual task.

## 1. Introduction

Falls are one of the most common reasons for immobilization and death in the older population [1]. Falls will be even more essential to address when large birth cohorts grow older and life expectancy increases. The number of older persons is rising and 16% of the total world population is expected to be 65 or older by 2050 [2]. Approximately 28–35% of people 65 years or older fall annually and the number of fallers increases to 32–42% in persons 70 years or older [1]. 

Falls have more severe consequences for older persons than younger, and fall-related injuries more often lead to death and serious injury among older persons [3]. For example, fall-related injuries such as hip fractures, upper limb injuries, and brain damages require medical care [1]. The expense for treatment is high and the reduction in functional ability after a fall may lead to permanently lower physical ability and lower level of independence [4]. In Sweden, there were 226 registered falls per 100,000 individuals over 65 years of age that directly led to death in 2015 [5]. A fall risk assessment and protective actions have been recommended to be considered, at the latest, after the first fall [6]. 

Gait disorder is regarded as one of the most relevant risk factors for falls [7,8]. The protective strategies commonly used among healthy older persons are characterized by: stance and gait base being widened, double stance phase is prolonged, step length becomes shorter, feet are lifted less high during swing phases, walking speed becomes slower, and posture becomes stooped. Once one of the above protective strategies fail while walking, a fall easily occurs [7,8]. Older fallers and non-fallers have different gait patterns, i.e., fallers have tendencies of decreased gait velocity and swing phase, as well as prolonged stance phase [9]. Furthermore, there seems to be a positive correlation between the variability of step time and the risk of falls, and step time variability can predict falls [10]. Step time variability is also positively correlated to strength and balance as well as gait speed [10]. 

Many activities of daily living demand the ability of multitasking such as walking outside and paying attention to the environment and traffic, and situations that require dual-tasking may endanger falls when the balance is decreased [6]. The most common reason behind falls is incorrect weight shifting, often during standing up, walking, or sitting down, and thus, a falls risk assessment should therefore include a multitask approach [11]. Changes in gait pattern when performing a dual-tasking exercise have a stronger association with the risk of future falls than when performing a single task exercise [12]. Furthermore, the variation in gait pattern while completing a dual-tasking exercise correlated positively with the risk of falls during a one-year follow-up [13]. 

It is time-consuming to assess gait parameters clinically and to monitor them over a long period. With modern technology, it is possible to measure gait parameters conveniently. However, whether modern technology, in the form of a wearable inertial measurement unit (IMU) sensor on the leg, can detect differences in gait parameters between fallers and non-fallers has not been studied clinically. 

The aim of this study was, therefore, to measure gait parameters among a group of older persons and to compare if there were any differences in gait parameters, vestibular function, and balance performance between fallers and non-fallers.

## 2. Materials and Methods

This was a longitudinal study with baseline tests and a 12-month follow-up.

### 2.1. Settings

The data collection was performed as a part of three balance-themed workshops for senior members in a non-profit gymnastic association in Malmö, Sweden during February–May 2017. The non-profit exercise center offers a variety of group exercise lessons and gym training. The recruitment of the participants was made using information posters at the center, and the interested persons signed up for the workshops. When signing up, the participants were given written information about the study. The recruitment was continued until the three groups were filled. 

Ahead of the workshops, an obstacle course was set up, which contained five tasks that included balance challenges often occurring in daily life. The five tasks were: standing up from a chair (41 cm high), walking within a narrow strip (25 cm in width) during a distance of 3 m, walking on an uneven surface, walking over three 30 cm high obstacles, and walking up and down stairs (steps in the stairs 17 cm high) (Figure 1). 

### 2.2. Participants

The participants were community-dwelling older persons who signed up for one of the three balance-themed workshops. Inclusion for participating in the study was being 60 years or older and considering themselves to be healthy. No further inclusion or exclusion criteria were used. The participants were asked for baseline information such as age, gender, illnesses, and medication. 

### 2.3. Baseline Measures

During the workshops, baseline outcome measures were performed. Gait parameters, in the form of average stride time, variation in average stride time, and gait flexibility were measured by a wearable IMU sensor, called the Snubblometer^®^ (“snubbla” is “stumble” in Swedish) (www.infonomy.com, accessed on 28 June 2021) attached to the right leg, 10 cm above the knee when participants walked the obstacle course. The IMU measures the accelerations during movements to calculate the time of each stride. The used unit of measurement for acceleration was meter divided by the square of a second (m/s^2^), and that of the stride time was millisecond (ms). An automated feature analysis of the IMU output was performed to isolate individual strides. Strides thought to be non-representative of the participant’s gait (i.e., the first stride after the participant had been standing still) were not included in the analysis. The IMU device was attached 10 cm above the knee with a thin strap. 

The participants were instructed to walk along the obstacle course at a comfortable speed. After traversing the obstacle course, the participant was instructed to walk back to the point of origin, with no obstacles and without any further details (Figure 1). The acceleration profile was sampled at 40 Hz.

Figure 2 displays a sample of data during one participant’s unobstructed walk. In post-processing, a simple step detection algorithm was applied, based on primary and secondary impact force, low-g detection during the terminal swing as well as standard time windows for these features in relation to each other. Standard threshold values were found for these parameters, and then manually tweaked in outlier cases to ensure that steps were always accurately detected.

Each step from the last section was resampled into 40 equidistant points, and the interpolated value at each point was stored in one of the 40 bins. By taking a median value of every bin, it was possible to construct a “mean step profile” for the unobstructed gait. Note that because of the resampling of the original data into a fixed number of bins per step, different bins in this acceleration profile represent the different phases of the gait cycle regardless of whether the user speeds up or slows down their gait. See Figure 2 for an overlay of this mean step profile on the raw acceleration data.

When performing the obstacle course, the participants were instructed to pause between each obstacle which allowed the data to be divided into different sections. The last section of the process, the walk back to the point of origin, was used as a benchmark of the user’s “normal” or “unimpeded” gait. Figure 3 contains an overlay of the “mean step profile” for the unobstructed gait on a series of steps from the obstacle course. It can be seen how some steps correspond well to an unimpeded step, while other steps deviate more.

For the sections corresponding to the obstacle course, i.e., all sections leading up to the last one, the same resampling algorithm was applied to each step, and the value in each of the 40 bins was compared to that of the calculated mean step profile. An absolute value was used to represent the relative deviation between bin *n* of the step in question and bin *n* of the mean step profile, and these values were averaged for all 40 bins. A percentage was thus acquired, representing how different the step in question was from the mean step during unobstructed gait. A value of 0% meant that all the re-sampled points corresponded perfectly to the mean step during unobstructed gait, while a value of 100% meant that, on average, each re-sampled point deviated by a value similar to the bin value of the mean step. The average value of this deviation from the obstacle course was used as a measure of how different the participant’s gait was during the obstacle course compared to their unimpeded gait, i.e., a measure of the participant’s gait adaptability to external circumstances, denoted as gait flexibility.

Vestibular function was assessed by a specialist in neurotology (author M.M.) with the Head Shake test [14], Dix–Hallpike test [15] and Head Impulse test [16], and manually reviewed. Balance was assessed by a physiotherapist (author E.V.) with the Timed Up and Go test (TUG), Timed Up and Go test with a cognitive task (TUGcog), and Timed Up and Go test with a manual task (TUGman) [17].

The Headshake test was used to assess vestibular asymmetry. The tested person was lying supine, and the examiner shook his or her head from side to side for 15 s. Eye movements were recorded with Video Nystagmoscope from Synapsis, and the test was considered pathologic if nystagmus beats occurred. The Headshake test has been suggested to be a sensitive indicator for diagnosing vestibular asymmetry [18].

Dix–Hallpike test was used to examine benign paroxysmal positional vertigo (BPPV) for the posterior and anterior canal. The tested person sat on an examination table, head turned about 45 degrees to the left or right. The examiner laid the patient down in a supine position, his or her head hanging back about 30 degrees. A positive test provoked nystagmus and vertigo [15]. The Dix–Hallpike has strongly been recommended to be used in diagnosing BPPV [19]. All participants who had a positive Dix–Hallpike test were recommended to contact their health care center for treatment. 

Head impulse test was used to examine hypofunction in the angular vestibular-ocular reflex. The tested person was asked to focus the gaze on the examiner’s nose, and the examiner rotated the patient’s head slowly from side to side. The examiner then turned the head rapidly about 20 degrees to one side. The test was repeated randomly to each side. A test result was considered pathologic if the tested person was not able to keep his/her eyes focused on the target but responded with a corrective or compensatory saccade [16].

Balance was assessed using Timed up and Go (TUG), Timed up and Go and doing a manual task (TUGman), and Timed up and Go and doing a cognitive task (TUGcogn). In TUG, the tested person was instructed to stand up from a chair with armrests, walk at a fast but safe pace three meters and cross a line marked on the floor, turn around, walk back to the chair and sit down [20]. The TUG test has been recommended by American and British Geriatric Societies as a screening instrument for falls [21].

In TUGman, the person proceeded as in the TUG test but after standing up, grabbed a glass filled with water from a table placed beside the chair and put the water glass back on the table after walking but before sitting down [20]. 

In TUGcogn, the person proceeded as in the TUG test but simultaneously counting aloud numbers down from 100 with 3-number intervals. TUGcogn has been stated to be a valid instrument for assessing future falls in elderly community-dwelling persons [20].

### 2.4. Falls

Falls were self-reported using a fall accident booklet, where the participants noted if a fall occurred and also described the fall. A new diary together with a pre-paid envelope was sent home to the participants every third month. Falls were classified using the Older Adult Service and Information System [22], where falls are classified into four different categories, extrinsic falls, intrinsic falls, non-bipedal falls, and non-classifiable falls. 

### 2.5. Statistics

Since the participants chose different strategies to walk down the stairs in the obstacle course, gait parameters for the total obstacle course (p2) as well as the obstacle course without walking down the stairs (p1) were calculated. Some of the age, gait parameters, and TUG test data sets were not normally distributed, and therefore, the non-parametric Mann–Whitney test was used to calculate differences between fallers and non-fallers. For calculating differences in proportions of participants with health issues and with vestibular dysfunction in the two groups, the Pearson’s Chi-square test was used. 

Since step time measured with a wearable sensor nor variance in step time have been used before to identify fallers, we used step time for power calculation. When counting 50% more falls among the persons who show a ≥2.0 ms difference in step time from the mean step time, an amount of 60 tested persons would be needed to reach a statistical significance of 0.05 [23]. Totally, 150 persons signed up for the workshop and were tested. Of these, 113 persons accepted to participate in the study. After excluding participants with incomplete baseline measurements and participants who did not return any fall diary, a total number of 101 participants were included in the statistical analysis. 

### 2.6. Ethical Considerations

Those who accepted to participate in the study gave their informed and written consent. They were informed that they, at any time, could withdraw from the study without further explanation and were assured confidentiality. All participants at the workshops (those who accepted to participate in the study and those who did not) were informed about their balance performance and their vestibular function. Those who had abnormal vestibular function were recommended to contact primary health care. The study was approved by the Ethical Review Board in Lund (2016/585).

## 3. Results

### 3.1. Participants

The study included 101 participants. The participants were 64–89 years old (mean 75.0 and SD 5.6) and the majority were female *n* = 91 (90%). Several of the participants did not declare any illnesses (*n* = 41, 40%) but almost half of these (*n* = 20) used medication, for example, for high blood pressure. Consequently, the classification of illnesses was based on both the illnesses and the medication that had been declared. Therefore, *n* = 28 (28%) of the participants were stated to suffer from cardiovascular illnesses, *n* = 9 (9%) of musculoskeletal illnesses, and *n* = 23 (23%) persons suffered from other illnesses. A flow chart of the study is shown in Figure 4.

### 3.2. Baseline Measures

Descriptive statistics, gait parameters, balance tests, and results for health status and vestibular tests are shown in Table 1. Of the total study population, 85 participants (84%) had a vestibular dysfunction (Table 1). Variability in stride time in one participant is displayed in Figure 5.

### 3.3. 12-Month Follow-Up

The majority of participants returned all four diaries (84 out of 101), 13 participants returned 3 diaries, one participant returned two diaries and two participants returned one diary. Forty-two subjects (42%) fell during the 12-month follow-up period, 37 of them were women and 5 were men (*p* = 0.07) (Table 1). The total amount of falls during the period was 58, nine persons had fallen twice, and one person had fallen eight times. Falls were categorized into extrinsic falls, caused by perturbed stance, perturbed gait or external cause of loss of balance, as well as intrinsic falls, caused by vertigo, legs giving way, or loss of postural control. The majority of falls were categorized as extrinsic falls (77%), and almost a fifth as intrinsic falls (19%). There were a few falls categorized as not classifiable (4%). All falls were evenly distributed during the 12-month follow-up and no differences were seen according to the season of the year or differences in distribution according to season between extrinsic and intrinsic falls (Table 2). 

There were differences between fallers and non-fallers in gait flexibility, both when walking the obstacle course without walking down the stairs (phase 1 *p* < 0.001) as well as when walking the total obstacle course (*p* < 0.001), where fallers had more limited gait flexibility than non-fallers, see Figure 6. Both when excluding (P1) and including (P2) the stairs phase in the obstacle course analyses, the fallers rarely adjusted their gait pattern more than 13% compared to unobstructed walking, whereas a clear majority of the non-fallers utilized a more flexible gait pattern than 13% to address the obstacles safely. The difference between fallers and non-fallers in gait flexibility when walking on an even surface displayed a clear trend (*p* = 0.058). No other differences between fallers and non-fallers were seen (Table 1).

## 4. Discussion

This prospective, observational study with a 12-month follow-up shows that, among a group of older persons, fallers had more limited gait flexibility compared to non-fallers when walking in an obstacle course.

These findings are conflicting with other studies concerning the difference in gait parameters among fallers and non-fallers [9,10]. However, in previous studies, gait parameters were measured using different technologies than ours and only on an even surface, which can explain the difference. The measure of gait flexibility used in our study is the total average deviation (measured as a percentage) of the user’s gait in the obstacle course compared to their characteristic stride. Gait flexibility reflects a person’s ability to adapt their gait to varying surfaces and obstacles. The higher level of gait flexibility, the higher ability to adapt. In our study, participants who later fell had, on average, lower levels of gait flexibility compared to non-fallers and thus probably used a more controlled and focused way to accomplish the obstacle course compared to non-fallers. A subject who has an overly controlled and careful gait already on a flat surface will likely not change their gait style much when presented with an obstacle. A person who can walk unhindered is expected to adapt their gait style for each obstacle, leading to a higher average deviation from their characteristic step. Thus, gait flexibility can be a useful measure for identifying future fallers among older persons. However, as displayed in Figure 6, not all fallers had low gait flexibility values. This suggests that the causes for falls might be diverse and can be an information source to utilize when categorizing fall incidents.

A total of 42% of the study population had sustained one or more falls during the follow-up period, which is in line with other research showing that about 30% of persons age 65 and above and every second person 75 years and above fall each year [6]. It is also in line with data from the World Health Organization [3].

We found no difference between fallers and non-fallers in respect to vestibular function. This was surprising since vestibular dysfunction has been shown to predict falls [24] and gait variability as in a previous study which has shown relation to vestibular dysfunction [25]. However, 84% of the participants in our study had vestibular dysfunction, which is a much higher proportion than has previously been found among healthy older persons [26], among older persons with fall-related wrist fracture [27,28], and among older persons with multisensory dizziness [24]. The large proportion of participants with vestibular dysfunction might explain why no difference in vestibular function was found between fallers and non-fallers. Additionally, 19 participants had a positive Dix–Hallpike test, indicating BPPV, and were recommended to contact their health care center for treatment. However, we do not know how many of these were treated during the 12 months’ follow-up. Moreover, if stricter inclusion criteria would have been used—such as the participants not having experienced any balance problems in the past year—this probably would have changed the sample to be more in correspondence with other studies on older persons and vestibular dysfunction. However, this would also have affected our possibilities to attract enough participants to the workshops.

No difference between fallers and non-fallers was found in any of the three TUG tests or the differences between TUG vs. TUGman and TUG vs. TUGcog. These findings are also unexpected since TUG has shown to be able to predict falls among recurrent fallers and TUGcogn has been shown to be able to predict falls among older persons living at home [20]. In our study, both fallers and non-fallers were faster in performing TUG (8.7 and 8.3 s) than the suggested cut-off value by Kang et al. (15.96 s), which can explain the difference in findings [29]. However, in TUGcogn, the difference from the suggested cut-off by Hofheins and Mib is reversed, which makes this difference even more surprising [20]. However, similar results of the TUG test having a limited ability to predict falls were found in two meta-analyses [17,30], concluding that the TUG test’s ability to identify fallers from non-fallers in high-functioning older persons was low [30].

The tests of vestibular function and interpretation of the test results were performed by an experienced ENT-specialist (author M.M.) according to clinical practice, and it is therefore likely that the findings of vestibular dysfunction are both valid and reliable. The TUG testing was, in all three workshops, performed according to practice by experienced physiotherapists, of which one was present in all three workshops.

The participants of the study were members of the non-profit association Friskis&Svettis, which provides exercise groups to their members. They were therefore expected to be more active and motivated in physical activities than a randomly chosen sample of the target population. However, they were invited to a workshop about balance and we suspect that many of the participants who signed up for the workshop had balance problems, which might explain the very high amount of vestibular dysfunction in the group.

The dropout rate was low, only four participants did not send in any diary during the 12-month follow-up period and 84 sent in all four diaries. The diaries were collected every third month, which is a time frame used in previous studies [31,32].

Further analyses of gait parameters when walking in an obstacle course with challenging balance tasks can elucidate if there is one or more specific task that stands out. From a clinical point of view, it will be useful to be able to use one or two tasks as a complement to a fall risk screening process or to identify persons at high risk of a fall.

## 5. Conclusions

This study showed that gait flexibility differentiates future fallers from non-fallers, where fallers had more limited gait flexibility when walking, both in the total obstacle course and when walking in the obstacle course without walking down the stairs, especially in a group with abundant vestibular impairments. In addition, the use of gait flexibility, captured by an IMU, seems better for identifying future fallers among healthy older persons than Timed Up and Go or Timed Up and Go combined with a cognitive or manual task.

## Figures and Tables

**Figure 1 ijerph-18-07074-f001:**
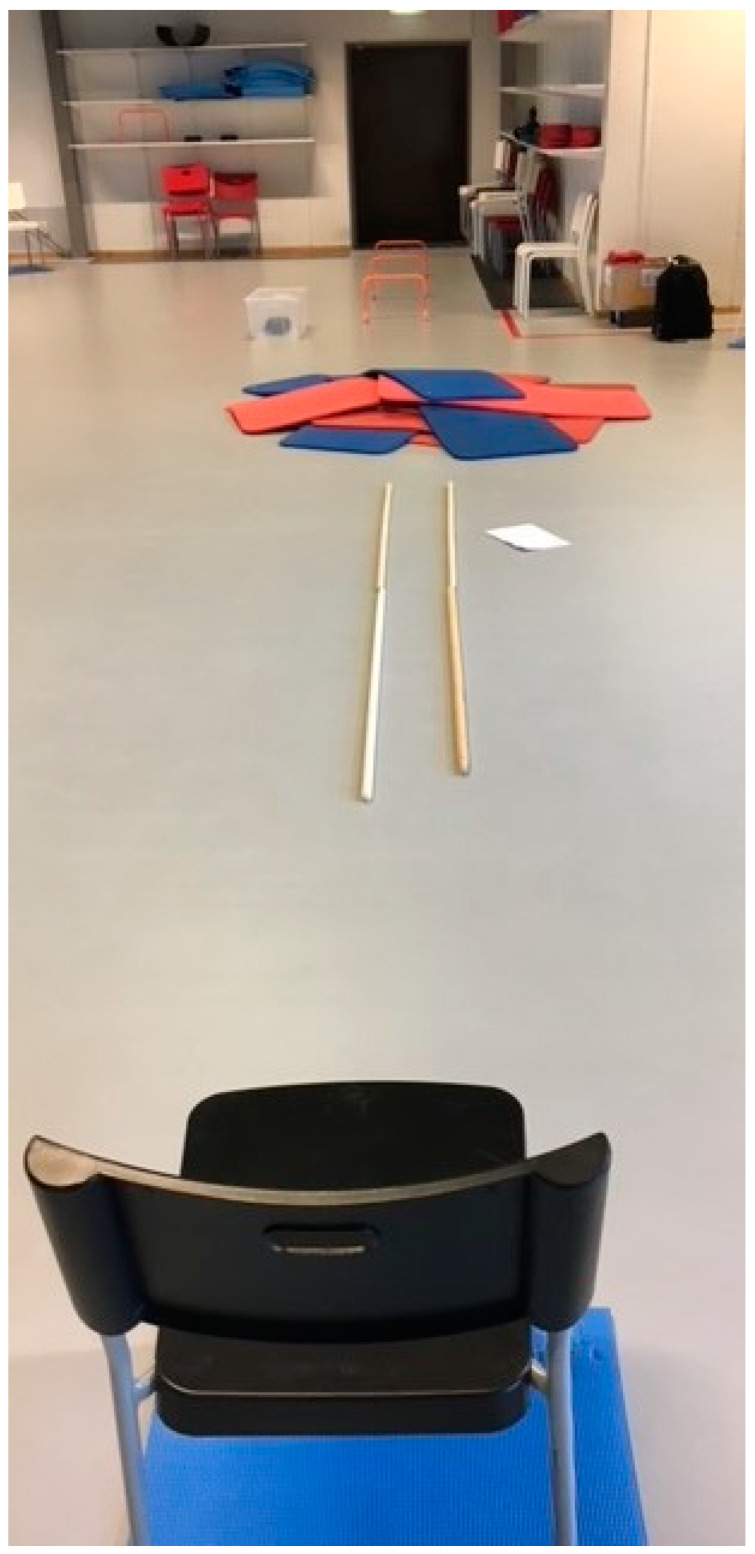
The obstacle course, including walking within a 25 cm narrow strip for 3 m, walking over an uneven surface, walking over three 30 cm high obstacles, walking up and down stairs with 10 steps (behind the door in the back of the picture), and walking back to the starting point.

**Figure 2 ijerph-18-07074-f002:**
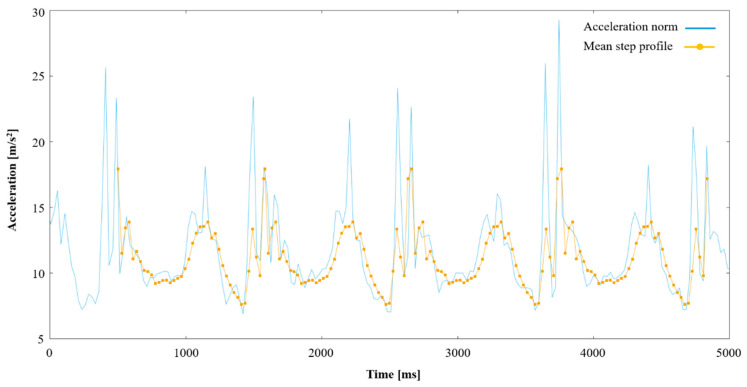
Acceleration norm during unobstructed walk presented together with an overlay of the calculated mean step profile.

**Figure 3 ijerph-18-07074-f003:**
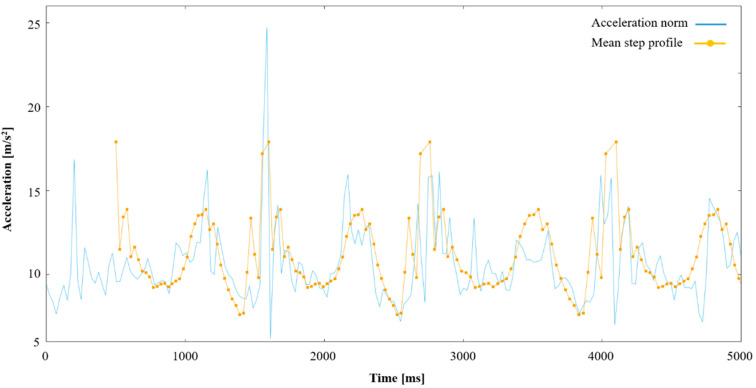
Acceleration norm during impeded gait (walking a part of the obstacle course) presented together with an overlay of the calculated mean step profile during unobstructed walk.

**Figure 4 ijerph-18-07074-f004:**
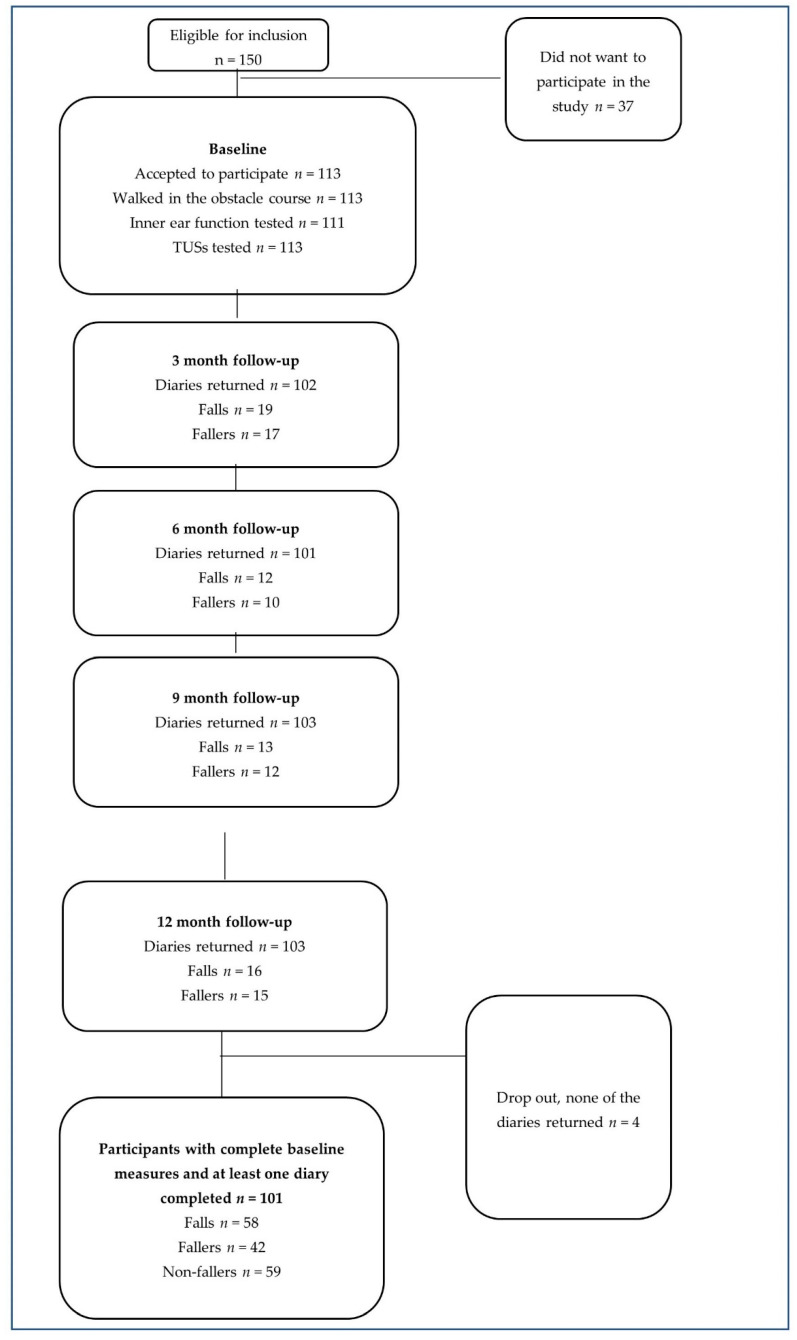
Flow chart of the study.

**Figure 5 ijerph-18-07074-f005:**
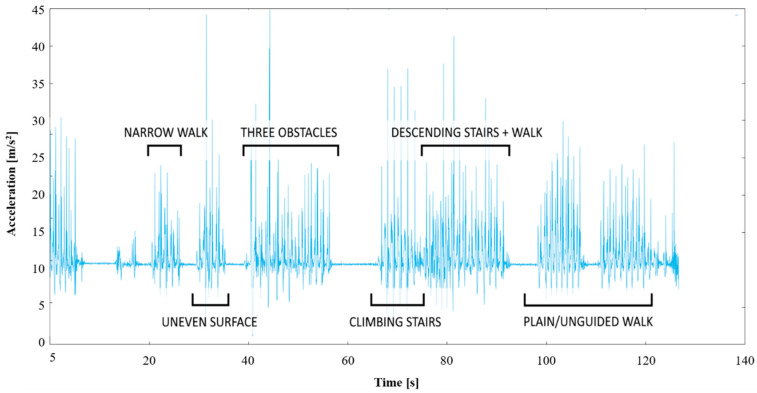
Variation in stride time for one participant while walking in the obstacle course.

**Figure 6 ijerph-18-07074-f006:**
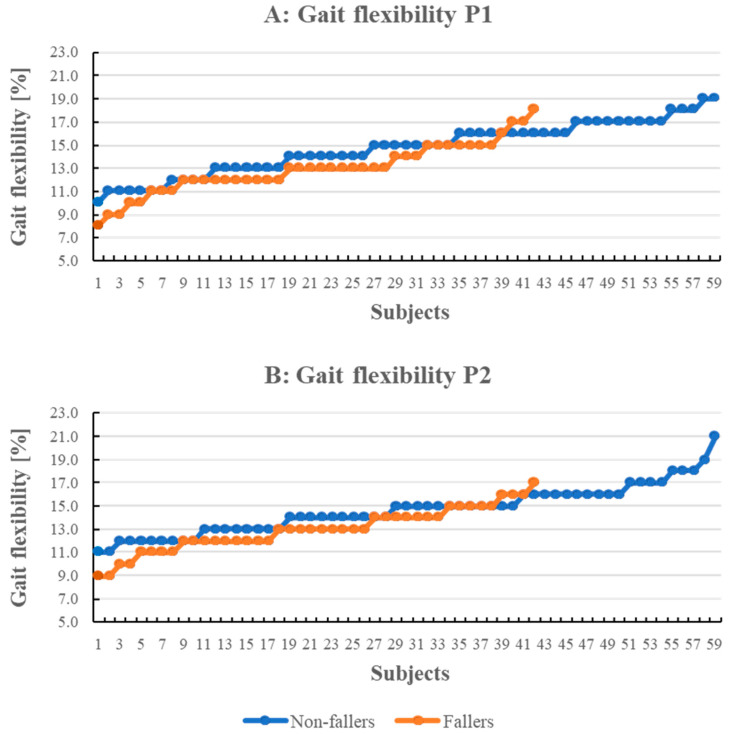
Gait flexibility for (**A**) P1 and (**B**) P2 obstacle courses. P1 = walking the obstacle course without walking down the stairs; P2 = walking in the total obstacle course.

**Table 1 ijerph-18-07074-t001:** Descriptive statistics of the participants, test results, and falls.

Evaluated Characteristics	*n* = 101	Fallers *n* = 42	Non-Fallers *n* = 59	*p*-Value
Age in years (mean (SD))	75.0 (5.6)	75.7 (6.2)	74.6 (5.2)	0.328
Health status (healthy/reported illnesses)	41/60	18/24	23/36	0.696
Gender number: Male/Female	10/91	5/37	5/54	0.404
Vestibular function (healthy/pathological)	16/85	6/36	10/49	0.788
Headshake test	24/77	11/31	13/46	0.629
Dix–Hallpike test	19/82	33/9	49/10	0.570
Head impulse test *^a^*	60/39	26/15	34/24	0.631
Total number of falls	58	58	0	na
**Gait parameters obstacle course (mean (SD))**				
Average stride time P1 *^b^* [ms]	1318.5 (100.8)	1319.0 (97.6)	1318.1 (103.9)	0.754
Variation in average stride time P1 [ms]	297.9 (72.9)	282.1 (75.6)	309.1 (69.4)	0.081
Gait flexibility P1 [%]	14.0 (2.4)	13.0 (2.2)	14.7 (2.3)	<0.001
Average stride time P2 *^c^* [ms]	1294.9 (88.1)	1296.2 (86.8)	1294.0 (89.8)	0.918
Variation in average stride time P2 [ms]	301.1 (66.5)	289.44 (65.8)	309.3 (66.3)	0.081
Gait flexibility P2 [%]	14.0 (2.1)	13.0 (1.9)	14.6 (2.0)	<0.001
**Gait parameters walking on an even surface (mean (SD))**				
Average stride time [ms]	1151.1 (80.5)	1168.2 (80.0)	1139.0 (79.2))	0.104
Variation in average stride time [ms]	187.9 (90.0)	190.6 (94.0)	185.9 (87.9)	0.793
Gait flexibility [%]	9.5 (2.5)	9.0 (2.3)	9.9 (2.5)	0.058
**TUG tests (mean (SD))**				
TUG [s]	8.4 (1.4)	8.7 (1.5)	8.3 (1.2)	0.308
TUGman *^d^* [s]	9.4 (2.1)	9.7 (2.2)	9.2 (2.0)	0.144
TUGcogn *^e^* [s]	10.2 (1.8)	10.6 (2.1)	10.0 (1.7)	0.157
diffTUGman *^f^* [s]	1.0 (1.4)	1.1 (1.3)	0.9 (1.4)	0.361
diffTUGcogn *^g^* [s]	1.8 (1.1)	1.9 (1.3)	1.7 (1.0)	0.469

*^a^* Data missing from two subjects; *^b^* P1 = walking the obstacle course without walking down the stairs; *^c^* P2 = walking in the total obstacle course; *^d^* Timed Up and Go with a manual task; *^e^* Timed Up and Go with a cognitive task; *^f^* Difference in time between TUG and TUGman; *^g^* Difference in time between TUG and TUGcog.

**Table 2 ijerph-18-07074-t002:** Distribution of falls according to season and according to extrinsic/intrinsic falls.

	Extrinsic Falls *n* = 46	Intrinsic Falls *n* = 10	Non-Classifiable Falls *n* = 2
May–July	13	3	1
August–October	9	2	1
November–January	12	1	0
February–April	12	4	0

## Data Availability

Data can be available upon request to the authors.

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
