# Peer review of "Gait Flexibility among Older Persons Significantly More Impaired in Fallers Than Non-Fallers—A Longitudinal Study"

_ijerph, 2021, doi:10.3390/ijerph18137074_

Round 1

Reviewer 1 Report

Gait Flexibility among Older Persons Significantly More Impaired in Fallers than Non-Fallers—A Longitudinal Study

Summary

The study aims to measure and compare gait parameters, vestibular function, and balance performance between fallers and non-fallers among a group of older persons, furthemore to investigate if wearable sensor can detect  differences in gait parameters between the two different populations described above.

Overall comment

This is a very interesting study regarding a field of research very challenging at the moment: the prediction of falls in older persons.  As falls represent a common complication in older persons, any scientific product aimed at improving knowledge on the incidence, risk factors, and preventive measures is of relevance to the National Health System. The inclusion of instrumental assessment may provide a systematic and objective approach to detect accurate and subtle fall risk factors. The paper is well writing, protocol and statistical analysis are well described. However, there are a minor concern as detailed below.

Discussion

The partecipants 60-years or older were included in the study.   No further inclusion or

exclusion criteria were used.in this study. The participants were asked for baseline information such as age, gender, illnesses, and medication. But on the discussion you wrote: “the 84 % of the participants in our study had vestibular dysfunction, which is much higher proportion than has previously been found among healthy older persons. The large proportion of participants with vestibular dysfunction might explain why no difference in vestibular function was found between fallers and non-fallers.”

Is it possible that a more accurate screening would change the sample?

The possibility to introduce the inclusion and exclusion criteria in your protocol should be discussed

Author Response

Thank you for helping us improve our manuscript, please find our point-to-point response in the attached document.

Reviewer 2 Report

Overall the paper is well written and the methods are sound. Authors present clear evidence on the effective use of IMUs with an obstacle course to measure user’s ability to adapt their gait (gait flexibility) as a predictor of likelihood of future falls.

Presentation of the data could be significantly improved.

  • There is significant redundancy in description of the methods (methods used in the study are repeated several times).
  • Flow chart used in Figure 1 would be better replaced by a picture or diagram of the actual obstacle course setup.  
  • Data for norm in figure 1 is redundant with figure 2, therefore figure 1 is not needed.
  • Overlay of the mean step profile for more than one step in figure 3, as well as comment in caption as to the gait flexibility of the data, would be helpful.
  • Data from Tables 1 and 2, could be easily presented in 1 table with 4 columns (total avg, fallers, non-fallers, and significance).
  • Scatter plot of gait flexibility with labels / rows for fallers and non-fallers, would give a better indication of the sensitivity / specificity of the measure.

Author Response

Thank you for helping us improve our manuscript. You will fin our point-to-point response in the attached file.

Reviewer 3 Report

This study aims to measure gait parameters among a group of older persons and to compare gait parameters, vestibular function, and balance performance between fallers and non-fallers. The work is interesting and the proposed approach is new. However, experiments must be better detailed for a better understanding, but also make them reproducible.

In the following my comments:

Please check the necessity to have the two keywords “balance” and “postural balance” at the same time. In the Introduction section, the concept “gait flexibility” is unclear for me. What’s the relevance between gait flexibility and gait disorder? In the settings section, how many uneven surface? How high is the each stair? In the participants section, the authors do not indicate the inclusion criteria that they have followed to choose that sample, of such different ages that it is not representative of any population or age. The need to indicate the sample calculation representation, for a study of this type, which cannot be carried out with a convenience sample, since they include a wide range of ages. In Line 219, SD is 5.8 but SD is 5.6 in Table 1. In Line 225, the figure 2 is not the flow chart. In Line 233, is the “figure 3” wrong, should be “figure 6”? In the Table 1, the “Health status” 51 + 60 equals to 111? But N = 101? Total number of falls is “58” or “60” in the figure 5? Fallers is 42 or 41? Non-fallers is 59 or 60? What’s the differences between healthy and sick? Healthy people are used as a control group, but how could it be said that their postural balance is normal? Because as previously mentioned, no statement was used prior to recruitment. At last, sick people are used as an experimental group, but it is unknown if their postural balance is adequate or pathological.

Author Response

Thank you for helping us improve our manuscript. You will find our point-to-point response in the attached file.

Round 2

Reviewer 3 Report

In Section 3.3 Line 392, the authors should explain the more information about figure 6. Please provide more gait flexibility (%) information between fallers and non-fallers in the paragraph. If possible, it would be better that the authors could provide more information about the meanings of overlapped curves in the discussion section. 

Author Response

Once again, thank you for helping us improve our manuscript. Please find our revised manuscript and point-to-point answer attached.
